# Multi-Material 3D Printing of Biobased Epoxy Resins

**DOI:** 10.3390/polym16111510

**Published:** 2024-05-27

**Authors:** Matteo Bergoglio, Elisabeth Rossegger, Sandra Schlögl, Thomas Griesser, Christoph Waly, Florian Arbeiter, Marco Sangermano

**Affiliations:** 1Politecnico di Torino, Department of Applied Science and Technology, Corso Duca degli Abruzzi 24, 10129 Torino, Italy; mattreo.bergoglio@polito.it; 2Polymer Competence Center Leoben GmbH, Sauraugasse 1, 8700 Leoben, Austria; elisabeth.rossegger@pccl.at (E.R.); sandra.schloegl@pccl.at (S.S.); 3Institute of Chemistry of Polymeric Materials, Montanuniversitaet Leoben, Otto Glöckel-Straße 2, 8700 Leoben, Austria; thomas.griesser@unileoben.ac.at; 4Materials Science and Testing of Polymers, Montanuniversitaet Leoben, Otto Gloeckel-Strasse 2, 8700 Leoben, Austria; christoph.waly@unileoben.ac.at (C.W.); florian.arbeiter@unileoebn.ac.at (F.A.)

**Keywords:** 3D printing, photopolymers, epoxy thermosets, multi-materials, biobased

## Abstract

Additive manufacturing (AM) has revolutionised the manufacturing industry, offering versatile capabilities for creating complex geometries directly from a digital design. Among the various 3D printing methods for polymers, vat photopolymerisation combines photochemistry and 3D printing. Despite the fact that single-epoxy 3D printing has been explored, the fabrication of multi-material bioderived epoxy thermosets remains unexplored. This study introduces the feasibility and potential of multi-material 3D printing by means of a dual-vat Digital Light Processing (DLP) technology, focusing on bioderived epoxy resins such as ELO (epoxidized linseed oil) and DGEVA (vanillin alcohol diglycidyl ether). By integrating different materials with different mechanical properties into one sample, this approach enhances sustainability and offers versatility for different applications. Through experimental characterisation, including mechanical and thermal analysis, the study demonstrates the ability to produce structures composed of different materials with tailored mechanical properties and shapes that change on demand. The findings underscore the promising technology of dual-vat DLP technology applied to sustainable bioderived epoxy monomers, allowing sustainable material production and complex structure fabrication.

## 1. Introduction

Additive manufacturing (AM), commonly known as 3D printing, has experienced significant growth in popularity over the past decade, serving as a versatile technology for rapid prototypes and material production in academia and industry. Unlike traditional manufacturing methods such as milling and moulding, AM allows the direct creation of complex shapes from digital CAD models, offering flexibility and efficiency [1,2,3,4,5,6,7,8].

When developing or modifying components, traditional manufacturing processes often require extensive changes, including the use of new tools or moulds. In contrast, 3D printing enables easy adaptation of the new component without physical changes or equipment, resulting in reduced material waste and cost savings. Both thermoplastic and thermosetting polymer materials can be processed by 3D printing.

Among the various 3D printing methods for polymers, vat photopolymerisation combines photochemistry and 3D printing. Digital light processing (DLP) 3D printing is a process within the vat-photopolymerisation techniques that employs a digital micromirror device (DMD) to control and direct UV light having a wavelength ranging from 380 to 450 nm. The light enables the photocuring of a photo-curable resin, resulting in high resolution and an optimal surface finish for the final pattern. The process involves the printing of a 2D slice of a CAD model, with the layer-by-layer curing process repeated while vertically moving the platform along the z-axis to create the desired three-dimensional form [9,10,11,12,13,14,15].

While acrylate monomers are among the most exploited precursors in the vat 3D printing process, Refs. [16,17,18,19,20] epoxy resins have also attracted interest in the field [21,22,23,24,25,26], mainly because the epoxy-based 3D printing structures are characterised by low shrinkage as well as high modulus and thermomechanical properties [27,28,29].

In a previous paper, we investigated the importance of coupling the use of chain transfer agents with an elevated temperature to enhance the reactivity of bis-phenol-A-epoxy resin in order to make them suitable for 3D printing technologies [30]. In this way, we were able to achieve 3D-printed structures characterised by high thermomechanical properties. However, it is important to take into consideration that conventional epoxy resins consist of the diglycidyl ether of bisphenol-A (DGEBA), which is cured in ambient conditions with polyamines or polyamides [31]. DGEBA is the condensation product of bisphenol-A (BPA) and epichlorohydrin. BPA is a reprotoxic R2 substance that negatively impacts wildlife and can lead to the destruction of the human endocrine system by mimicking the human body’s hormones [32,33].

The toxicity of the most commonly used epoxy resins, including their precursors, such as BPA, raises the demand to research alternatives. Bioderived epoxy resins, including epoxidized linseed oil (ELO) and vanillin alcohol diglycidyl ether (DGEVA), can be considered promising alternatives to conventional thermosets. The research concerning these materials follows the green deal for a sustainable transition, offering environmental benefits [34,35]. For this reason, bioderived epoxy resins have been the subject of previous studies in our research group [36,37,38,39,40,41,42,43]. Also, biobased epoxy precursors were investigated in 3D printing technologies. Furandimethanol diglycidyl ether (FDE) and resorcinol diglycidyl ether (RDE) were investigated via the Hot Lithography technique [44]. Both monomers are obtained from the conversion of C-6 monosaccharides; the FDE is derived from hydroxymethylfurfural (HMF), which is derived from vegetable biomass carbohydrates such as glucose and fructose, while RDE can be produced starting from glucose. The biobased precursors were functionalized on purpose with epoxy rings and exploited in 3D printing. 

However, while 3D printing of single materials using epoxy monomers has been investigated, the fabrication of multi-material bioderived epoxy thermosets remains unexplored to the best of our knowledge. Multi-material 3D printing offers the possibility of integrating materials with different properties. This capability enables potentially the manufacture of complex structures for various applications such as drug delivery [45], tissue engineering [46] soft robotics [47] or electronics [48].

In vat photopolymerisation 3D printing, multi-material properties can be realised by adjusting the intensity of the UV lamp during printing (also known as greyscale printing), changing the colour/wavelength of the UV lamp or by changing the vat [18,49,50,51,52]. In contrast to other multi-material techniques, multiple-vat printing offers greater freedom in network design because it allows the selection of any photocurable resin and the combination to build an object containing different sections. Additionally, unlike greyscale printing, the resulting objects are not prone to migration and post-polymerisation reactions of a high number of unreacted species, which can negatively affect the post-printing performance of multi-material parts.

In this work, we applied the concept of dual-vat DLP 3D printing, which involves the automatic exchange of vats for the multi-material printing of bioderived epoxy resins. Multi-layer objects and specimens were printed with different mechanical characteristics by utilising epoxidized linseed oil (ELO) and vanillin alcohol diglycidyl ether (DGEVA) bioderived epoxy monomers. The curing process was investigated, and the 3D-printed object was fully characterised, demonstrating the possibility of achieving multi-layer epoxy thermosets capable of exhibiting on-demand properties.

## 2. Materials and Methods

### 2.1. Materials

Epoxidized linseed oil (ELO) was purchased from HOBUM OLEO Chemicals (Hamburg, Germany), vanillin alcohol diglycidyl ether (DGEVA) was obtained from Specific Polymers (Castries, France), and the photoinitiator p(octyloxyphenyl) phenyliodonium hexafluoroantimoniate (photoinitiator) was purchased from Gelest (Morrisville, NC, USA). Isopropylthioxanthone (ITX) and 1-(2,4-xylylazo)-2-naphthol (Sudan II) were obtained from Sigma-Aldrich (Milano, Italy). All the products were used without further modification.

### 2.2. Preparation of Photocured Formulations

Two distinct formulations, designed for printing in the different vats, were prepared using the selected resin, ELO and DGEVA, respectively. Both formulations included 2 phr (parts per hundred parts) of photoinitiator, 0.02 phr of Sudan II, and a variable amount of the photosensitiser ITX. The detailed composition of the prepared resin is provided in Table 1.

The photo-adsorber (Sudan II) addition aimed to prevent excessive photopolymerization, thereby enhancing resolution during the DLP 3D printing process. All four components in each formulation were combined and stirred together in an ultrasonic bath for 40 min at 50 °C to ensure the complete dissolution of Sudan II in the resin. DGEVA, which is solid at room temperature, was preheated above 60 °C for 30 min to facilitate the melting of the crystalline phase and achieve a sufficiently low viscosity for effective mixing. Resin preparation and storage were conducted in light-protected glass vials. The chemical structures of the components are reported in Figure 1.

### 2.3. Characterisation Methods

#### 2.3.1. Photo Dynamic Scanning Calorimetry (Photo-DSC)

The crosslinking reaction was monitored using a NETZSCH photo-DSC 204 F1 Phoenix (Selb, Germany) instrument equipped with an Omnicure s1500 UV-light source (Feldkirchen, Germany). The measurements were performed under a nitrogen flow of 40 mL/min. 

#### 2.3.2. Attenuated Total Reflectance Fourier Transform Infrared Spectroscopy (ATR-FTIR)

The curing reaction was monitored using a Bruker Vertex 70 FT-IR (Milano, Italy) spectrometer in transmittance mode with a Si substrate. The sample, spread evenly over a 40 µm thickness on the Si substrate, was analysed with a spectra resolution set to 4 cm^−1^ and 16 scans. The curing process was followed by observing the disappearance of the epoxy group situated around 800 cm^−1^. The following Equation (1) was applied throughout the exposure time to determine the conversion behaviour.
(1)$$Conversion\left(\%\right)=\frac{({\frac{A_{group}}{A_{ref}})}_{t=0}-({\frac{A_{group}}{A_{ref}})}_{t}}{({\frac{A_{group}}{A_{ref}})}_{t=0}}\times 100\%$$

Here, A_ref_ correspond to the area of the reference peak around 2900 cm^−1^, since the C-H stretch was considered unaffected by UV light, while A_group_ corresponds to the area of the epoxy group located around 800 cm^−1^.

### 2.4. Dual-Vat Digital Light Processing (DLP) 3D Printing

Multi-material 3D printing was conducted on a prototype printer manufactured by W2P Engineering GmbH (Wien, Austria). The printer had a building area measuring 67.2 × 37.8 × 130 mm, and each vat could contain a maximum of 210 mL of resin. The light engine was fixed in position, while the double vat had the capability to move along the x-axis on a linear sliding platform (Figure 2). The layer thickness was adjustable within the range of 25 to 200 µm. To enhance the fluidity of both epoxy resin and, in the case of resin D2, maintain the fluidity of DGEVA (which is solid at room temperature), the printing chamber was heated to 50 °C. The DLP printer was equipped with a light engine from In-Vision Digital Imaging Optics GmbH (Wien, Austria), operating at 405 nm with a maximum intensity of 40 mW/cm^2^ and 50 µm pixel size.

Both resins were printed using a 25 µm layer thickness, employing a rising and retracting speed of 5 mm/s. Resin E1 was printed using a bottom layer irradiation time of 20 s, followed by 15 s irradiation of each layer. D1 resin was printed using 30 s irradiation time for the bottom layer and 4 s for the subsequent layers. To complete the reaction and eliminate the uncured monomers, the multi-material epoxy thermoset was rinsed with isopropanol and post-cured for 30 min by using a 405 nm lamp.

#### 2.4.1. Single-Material DLP 3D Printing

DLP 3D printing of the epoxy resin was conducted using the optimised formulations E1 and D1. Single-material printing was conducted on the dual-VAT DLP prototype printer using only one vat. Single-material printing involves the use of a single VAT without employing the double-vat sliding mechanism. The printing process included the movement of the upper platform on the *z*-axis with a precision of 25 µm.

#### 2.4.2. Multi-Material DLP 3D Printing

Multi-material printing involved switching the vats that contained the different resins. During printing, the platform was automatically raised to the safety position in the *z*-direction (50 mm above the zero point) once the defined layer of the chosen material layer reached the desired thickness. The moving lower platform then exposed the cleaning station to the metal upper platform, which consisted of a sponge, in which the material was pressed inside with a 10 mm depth. When the material encountered the sponge, the lower platform moved 25 mm along the *x*-axis forward and back to completely remove the uncured liquid resin. This method is safer than the classical solvent methods, and the material properties are not at risk of alteration through contact with solvents. 

After cleaning, the platform was raised again to a safe distance (50 mm above the zero point), and the vat containing the second resin was exposed by automatically moving the lower platform along the *x*-axis. The process continued similarly until all the multi-material layers were completed.

### 2.5. Jacobs Working Curves 

The light-resin interaction was investigated by creating the Jacobs curve, also known as the working curve, for both resins E1 and D1, according to the literature [53]. In the experiment, one-layer discs of 10 mm in diameter were printed by varying the exposure time from 15 to 150 s while maintaining a constant light intensity of 30 mW/cm^2^. Following the printing process, all the discs were rinsed with isopropanol to remove any excess non-cured liquid resin and were subsequently post-cured for 30 min. The 3D printing of the samples was replicated twice, and the mean thickness value was measured using a thickness gauge. 

The measured thickness values were utilised to determine the working curve using the following Equation (2): (2)$$C_{d}=D_{p}\times ln(\frac{E_{max}}{E_{c}})$$
where C_d_ corresponds to the cure depth (measured as the circle thickness in micrometres), E_max_ (mJ/cm^2^) represents the incident light energy on the vat surface, and E_c_ (mJ/cm^2^) is the characteristic quantity specific to the resin, indicating the required exposure dose for the transition from liquid to solid state (gel point). Additionally, D_p_ is another specific characteristic resin parameter that indicates the penetration depth of the incident light (µm). A semilogarithmic plot of C_d_ vs. E_max_, derived from the working curve, is expected to form a straight line. The slope of the line corresponds to D_p_ at the wavelength of the light source used to generate the working curve. Theoretically, the cure depth corresponds to zero at E_c_, which represents the x-axis intercept of the working curve. The critical value E_c_ indicates the necessary energy to reach the gel point of the resin.

### 2.6. Mechanical and Thermomechanical Properties

Dynamic Mechanical Thermal analysis (DMTA) was performed using a Mettler Toledo DMASDTA861e (Wien, Austria) dynamic mechanical analyser. Rectangular samples, with a thickness of 0.5 mm and a width of 3.5 mm, were tested in displacement-controlled oscillation using an amplitude of 10 µm, a frequency of 1 Hz, and a clamping distance of 10.5 mm. Mechanical loss factor (tan δ) and storage moduli (E’) were measured over a temperature range between −20 and 140 °C, with a heating rate of 3 °C/min. 

Tensile tests were performed on a Zwick Z001 from ZwickRoell GmbH & Co. KG. (Ulm, Germany) equipped with a 1 kN load cell. A loading rate of 1 mm/min was used to determine the Young’s modulus, after which the loading rate was changed to 20 mm/min until specimen fracture. Dog-bone-shaped samples were 3D printed according to ISO 527-2 [54] type 1B normative. All tests, performed on an average of 5 samples for each formulation, were performed under ambient conditions. 

Flexural tests were performed on samples featuring rectangular cross-sections measuring 5 × 2 mm^2^ and a length of 20 mm. These tests were carried out utilising a Microtest bending stage manufactured by DEBEN UK Ltd. (Suffolk, UK), which was equipped with a 200 N load cell. The loading rate was maintained at 1 mm/min until a sample fracture occurred. The distance between the supports was set to 16 mm, with both the fin and the supports having radii of 2 mm. Stress, strain, and flexural modulus were determined following ISO 178 [55] on three replicates under ambient conditions.

Thermogravimetric analysis was performed by using a Mettler-Toledo TGA/DSC thermogravimetric analyser (Mettler Toledo, Wien, Austria) from 25 to 900 °C at a heating rate of 10 K/min under a nitrogen atmosphere in aluminium oxide crucibles. The measured curves were evaluated with the STAR^e^ Evaluation Software (star30) (Mettler-Toledo, Wien, Austria).

### 2.7. Shape Memory Tests

Shape memory tests were performed on a multi-material 3D-printed flower with dimensions of 37 × 37 × 0.2 mm. The flower, initially printed on a planar plane, was programmed after being placed into an oven at 80 °C for 15 min in a closed shape. Once the closed petal shape was obtained, the object was positioned onto a heating plate and heated up to 100 °C, and the shape change was monitored.

## 3. Results and Discussion

### 3.1. Photo-Curing Process

The photo-curing process of the selected formulations was monitored through photo-DSC and FTIR studies. Photo-DSC analyses were performed on the resin mixture to optimise the isopropylthioxanthone (ITX) concentration exploited as photosensitizer. The initial formulations, denoted as E1 and D1 (see Table 1), comprised the pristine biobased epoxy monomers (ELO and DGEVA, respectively), along with 2 phr of iodonium salt as a cationic photoinitiator and 0.02 phr of Sudan II (used as dye in the 3D printing process). These concentrations were selected based on previous studies on epoxy resin within our research group [40,43,56]. A first-step photo-DSC analysis was performed on the pristine formulation, without ITX, at 365 nm to evaluate the temperature that best enhances the resin reactivity. The exothermic curves are reported in Figure 3a,b, respectively, for ELO and DGEVA formulations.

It is evident from the photo-DSC curves that raising the temperature to 50 °C induced an important enhancement of the epoxy reactivity with an increase in exothermicity. Therefore, a temperature of 50 °C was chosen for the subsequent photo-DSC analysis to determine the optimal content of ITX as a photosensitizer. The photo-DSC curves, recorded by irradiating the mixtures at 405 nm wavelength (the same wavelength of emission of the printer), are reported in Figure 4a,b, respectively, for ELO and DGEVA formulations containing different contents of ITX. All the recorded data are collected in Table 2.

The curing process, as described by Crivello et al., follows a cationic photopolymerisation. When the photoinitiator is reached by UV light, it generates a Bronsted acid that initiates the cationic chain-growth polymerisation [57]. Analysing the photo-DSC data, it is evident that, in the case of the E1 formulation, ITX is necessary to allow crosslinking reactions when using a longer wavelength. In the absence of ITX, the heat released is close to zero. Conversely, when ITX is added to E1 resin at concentrations of 1 and 2 phr, the heat release curve increases rapidly, proving the occurrence of the reaction and the efficiency of the photosensitizer. The optimal value was determined to be 1 phr of ITX since a higher amount of ITX did not lead to a significant enhancement in heat released, suggesting that a large amount of ITX does not change the reaction kinetic. 

Similarly, the D1 formulation does not fully react immediately when irradiated at 405 nm in the absence of photosensitiser, and the optimal amount of the ITX corresponds to 1 phr since a higher concentration leads to a competitive absorption within the formulation. 

As a second step, FTIR analyses were performed on the optimised formulations. The results are reported in Figure 5 for E1 and D1 formulations containing 2 phr of cationic photoinitiator, 1 phr of ITX photosensitizer, and 0.02 phr of the dye Sudan II.

The conversion curves showed a higher reactivity and final epoxy group conversion for D1 with respect to E1 epoxy-based formulation. This agrees with photo-DSC data, showing the higher reactivity of DGEVA-based formulation.

### 3.2. Single-Material DLP 3D Printing

The single material printing process was performed at 50 °C, based on the preliminary investigations reported above. The optimal light irradiation time per layer was determined from the Jacobs working curves for the E1 and D1 formulations, reported in Figure 6. The irradiation resulted in 15 s for the E1 resin and 4 s for the D1 resin, obtaining a satisfactory printing definition, as visible in Figure 7. Tensile dog bone-shape samples, test printing objects (Figure 7), and DMTA samples were printed from both resins.

### 3.3. Multi-Material DLP 3D Printing

The printed multi-material shape included a DMTA double-layer and three-layer sample (4 × 20 × 1.5 mm), a double- and triple-layer dog bone-shaped tensile specimen, and a multi-material flower structure. The printed DMTA samples are reported in Figure 8.

### 3.4. Characterisation Tests 

#### 3.4.1. DMTA

Dynamic mechanical analysis was performed on all printed specimens, including the multi-materials. The results obtained by the analysis are reported in Figure 9 and Figure 10, and the data collected are shown in Table 3.

From the analysis, a notable difference between E1 and D1 was observed. The ELO-crosslinked material showed a lower stiffness (T_g_ around 50 °C) with respect to the D1 crosslinked networks (T_g_ around 96 °C), primarily due to the presence of the highly flexible triglyceride chains in ELO, while the stiffness of the crosslinked D1 resin can be due to the presence of the aromatic benzene structure in the DGEVA. A distinct behaviour is noticeable in the T_g_ of the multilayer material: for both printed double and triple-layer specimens, the glass transition temperature results are shifted towards the T_g_ value of the photocured formulation D1. Therefore, the stiffer layer of the specimen contributes to an overall increase in the glass transition temperature, suggesting an ideal incorporation of the soft material.

#### 3.4.2. Mechanical Tests

Tensile tests were performed on all multi-material 3D-printed dog bone-shaped specimens. The results are reported in Figure 11 (one representative curve per material or material combination) and summarised in Table 4.

The elastic modulus of the pristine light-cured epoxy materials agrees with the evaluated T_g_ values, showing a modulus of 505 MPa for the E1 polymeric network and a modulus of 1615 MPa for the D1 light-cured material, proving that an increase in the cross-linking density of D1 leads to a higher T_g_ and a higher modulus. The values (Young’s modulus and tensile strength) obtained for the multi-material specimen are located between the two extremes. The bi-material E1-D1 curve aligns closer to the stiff material, proving that the D1 layer contributes significantly to the structural properties that absorb the majority of the force during the test. In the three-layer specimen tests, the curves remain situated between the outer layers. The curve regarding the specimen containing a higher layer amount of D1 is positioned above the double-layer material, suggesting that the stiffer portion absorbs the overall stresses. However, the outcomes are as expected. The stiffer material, D1, contributes to the increase in strength and stiffness, while the softer components of E1 enhance the elongation at break.

In Figure 12, the results obtained from the flexural tests are illustrated (one representative curve per material or material combination). The results, encompassing stiffness, strength, and elongation at break, are summarised in Table 5. Despite the different stress states within the samples for bending and tensile, the curve profiles of the mono-materials and the multi-materials closely align with those observed in tensile tests. The pure UV Cured epoxy materials exhibited the highest or lowest stiffness and strength for D1 and E1, respectively. The multi-material samples were located between these two extremes. 

The decrease in strength and stiffness from D1 to E1 by using multi-materials can mainly be found in the phenomenon of stress decoupling compared to the tensile test data. Embedding a soft layer into a stiff matrix prevents significant stresses from being transmitted across this soft layer when it is subjected to bending. The stiffer matrix ligament behaves like a single bending beam [58,59]. This essentially explains why the mechanical properties of E1-D1 and E1-D1-E1 are very similar and significantly different from those of D1-E1-D1. The proportions of D1 enhance stiffness and strength but adversely affect elongation at break. 

In particular, the embedding of a soft interlayer into a stiff matrix, as in the case of D1-E1-D1, has been shown in various publications [60,61,62,63,64] to have a positive effect on toughness enhancement. However, this effect did not manifest here, likely due to an insufficient mismatch in mechanical properties. Nevertheless, in future endeavours, it may be possible to customise the mechanical properties to align with a specific desired mechanical profile by manipulating factors such as the number [59], arrangement [58], or thickness [64] of the layers.

#### 3.4.3. TGA

The thermal stability of the 3D-printed samples was determined by means of thermogravimetric analysis (TGA), focusing on single-material specimens. The results are illustrated in Figure 13. Both samples exhibit a single degradation step, wherein the DGEVA-based sample experiences degradation around 410 °C with the onset of thermal degradation at approximately 310 °C. On the other hand, the ELO-based resin demonstrates a maximum degradation rate at 400 °C and an onset of thermal degradation at 330 °C. The single step of degradation for both resins explains the degradation of the singular thermoset structure. The E1 degradation derivative, despite being similar to the D1 analogue, is broader, and that has to be attributed to the lower conversion of the D1 resin, which brings differences in the chain length of the UV-cured thermoset, hence a broader degradation derivative. On the other hand, since the D1 resin reached the highest conversion of the epoxy groups, it contains a linear structure that degrades faster. Overall, both formulations show high thermal degradation resistance.

### 3.5. Shape Memory

The printed flower structure, illustrated in Figure 11, was investigated as a shape memory structure. The 3D-printed object was heated up to 115 °C and then deformed into a closed flower shape. After deformation, the structure was cooled to room temperature to stabilise the programmed temporary shape. The chosen heating temperature was set above the two epoxy networks’ T_g_ but below the degradation temperature measured by TGA.

After the initial programming, the flower was placed on a heating plate, and its behaviour was observed when the temperature was raised (the video can be viewed in Appendix A). As expected, upon heating, different sections of the flowers recovered their initial planar shape at different temperatures. Above the first glass transition temperature (T > T_g1_) attributable to the lower T_g_ material E1, the softer petals printed with the E1 resin recovered their initial position, while the stiff petals maintained their curved shape. Once the temperature rises above D1 T_g_ (T > T_g2_), the other part of the object returns to its initial shape, obtaining once again the printed permanent structure. The different steps are reported in Figure 14. This process can be repeated various times and showed the shape memory behaviour of the 3D printed structure, which was achieved by exploiting the possibility to printing the object with two different materials characterised by a large temperature interval of T_g_.

## 4. Conclusions

This study investigated multi-material 3D printing using epoxy resins based on bio-derived monomers, epoxidised linseed oil (ELO), and vanillin alcohol diglycidyl ether (DGEVA). The dual-vat DLP 3D printer, equipped with two automatically interchangeable vats and a cleaning station, allowed the creation of multilayer objects that were deeply characterised. 

Initially, the characterisation focused on individual pristine materials for each formulation (E1 for ELO-based and D1 for DGEVA-based), followed by a detailed analysis of the multilayer objects. Preliminary studies conducted by means of photo-DSC and FTIR played a crucial role in determining the optimal composition of photocurable formulations in terms of photoinitiator and photosensitizer content and the ideal temperature for the occurrence of the cationic light-activated cross-linking reaction.

Dynamic mechanical thermal analysis showed a higher glass transition temperature (T_g_) for D1 crosslinked material compared to E1. Furthermore, DMTA analyses highlighted the synergistic behaviour of the multi-material specimens, revealing a glass transition temperature shifted towards a higher value in the multilayer structure, proving the positive effect of the multiple layers.

Mechanical tests revealed a superior Young and flexural modulus in D1 resin compared to E1. During the tensile tests, the multilayer specimens demonstrated properties close to D1 crosslinked material, indicating enhanced mechanical performance for multilayer materials. The flexural tests supported the tensile tests, proving that the D1 component increases strength and stiffness while the softer E1 component enhances the elongation at break. 

The shape memory capabilities of the 3D printed flower structure highlighted the innovative potential of multilayer printing, particularly in applications involving thermal responsiveness. The structures exhibited the ability to change shape on demand, activated by temperature. The properties make the material suitable for various industrial applications, such as automotive and aerospace, consumer electronics, innovative smart textiles, and revolutionary medical devices. The utilisation of multi-materials in this sector can bring new properties and functions to the already known excellent properties of epoxy thermosets, while also increasing their sustainability thanks to the biobased nature of the monomers. 

In essence, this study showcased the remarkable capabilities of dual-vat 3D printing using biobased epoxy resins. The versatility of the technique allows the printing of an extensive range of materials, with the flexibility to vary the chamber temperature and the UV-light intensity during the process. 

## Figures and Tables

**Figure 1 polymers-16-01510-f001:**
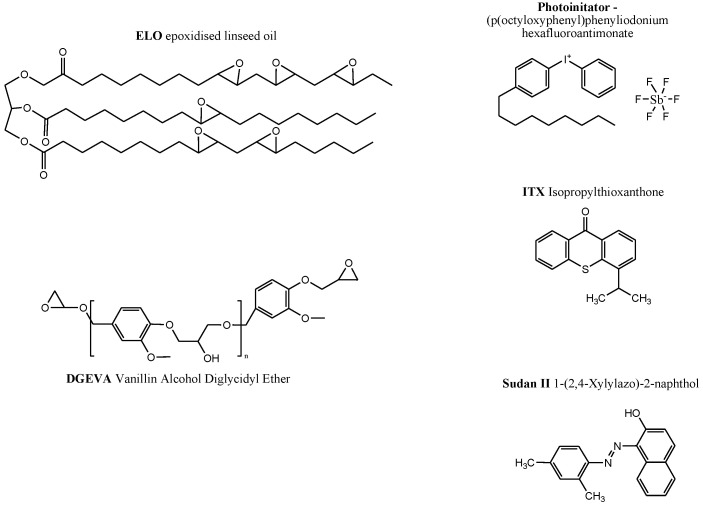
Chemical formula of biobased epoxy monomers and chemical compounds.

**Figure 2 polymers-16-01510-f002:**
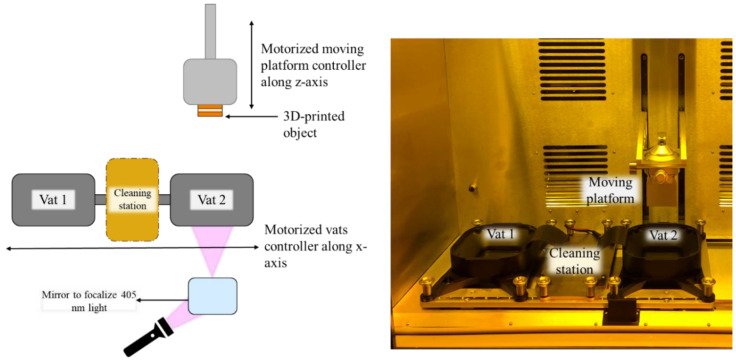
Scheme of the dual-vat 3D printer manufactured by W2P.

**Figure 3 polymers-16-01510-f003:**
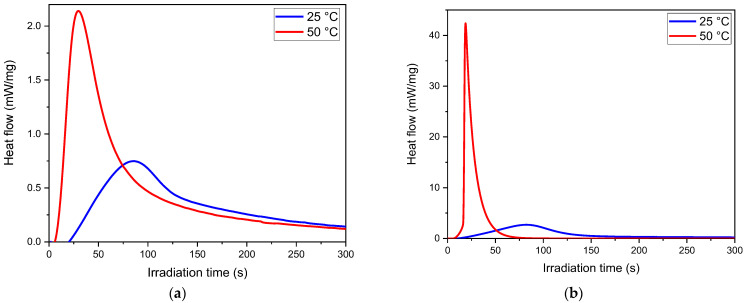
Photo-DSC curve of (**a**) E1 formulation (ELO based) and (**b**) D1 formulation (DGEVA based) performed at 25 and 50 °C. Irradiation at 365 nm wavelength.

**Figure 4 polymers-16-01510-f004:**
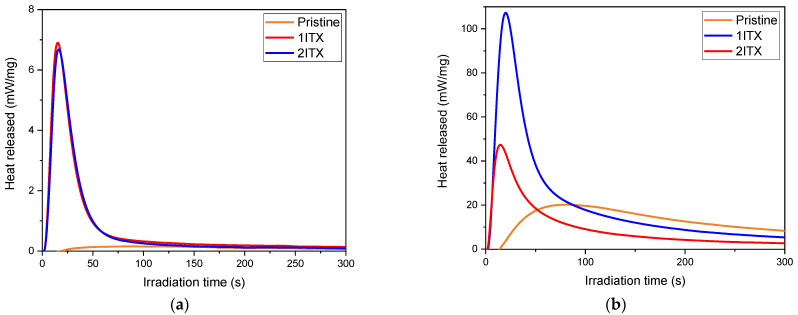
Photo-DSC curve of E1 (**a**) formulation (ELO based) and D1 (**b**) formulation (DGEVA based) with addition of ITX as photosensitizer, performed at 50 °C and irradiation at 405 nm wavelength.

**Figure 5 polymers-16-01510-f005:**
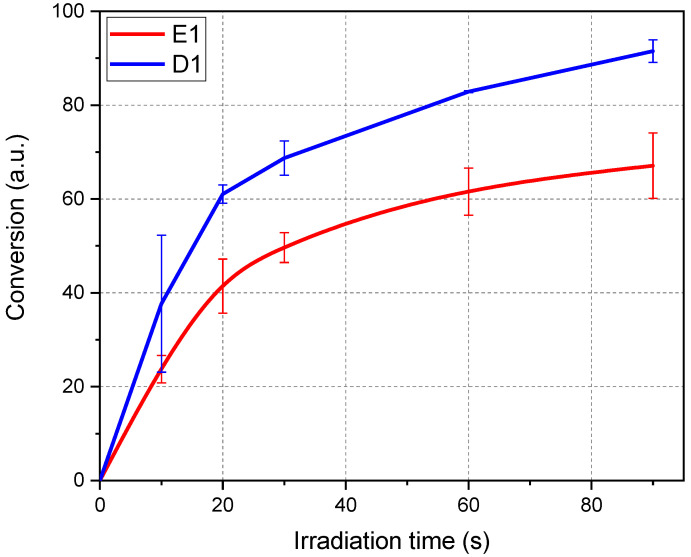
FTIR conversion of E1 and D1 resin photocurable formulations.

**Figure 6 polymers-16-01510-f006:**
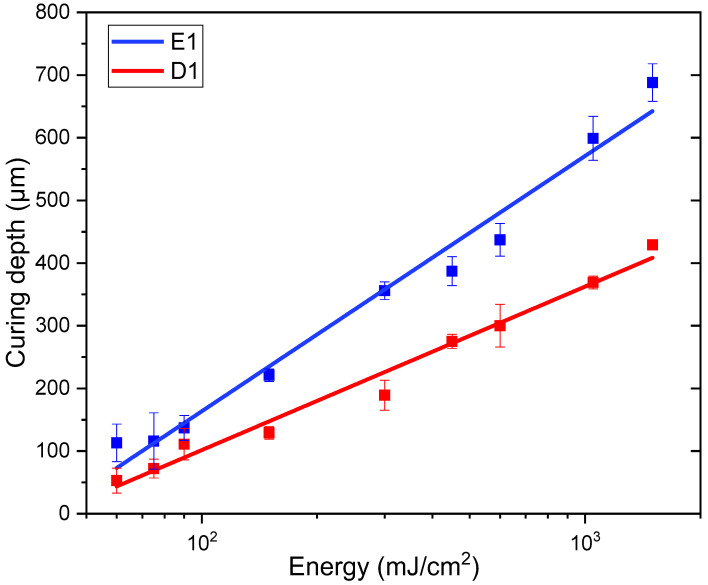
Jacobs working curves obtained from E1 and D1 resin.

**Figure 7 polymers-16-01510-f007:**
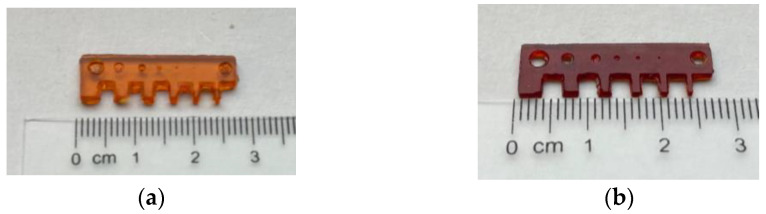
Definition printing object of E1 (**a**) and D1 (**b**) resin.

**Figure 8 polymers-16-01510-f008:**
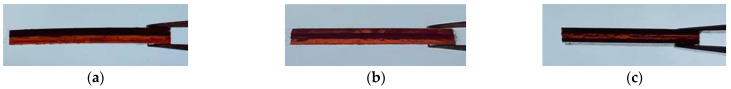
Double-layer DMTA sample (**a**), triple-layer E1-D1-E1 DMTA sample (**b**), triple-layer D1-E1-D1 DMTA sample (**c**).

**Figure 9 polymers-16-01510-f009:**
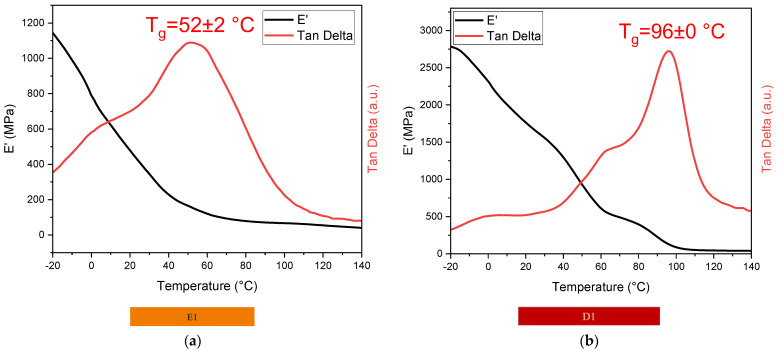
DMTA analyses of E1 (**a**) and D1 (**b**) samples.

**Figure 10 polymers-16-01510-f010:**
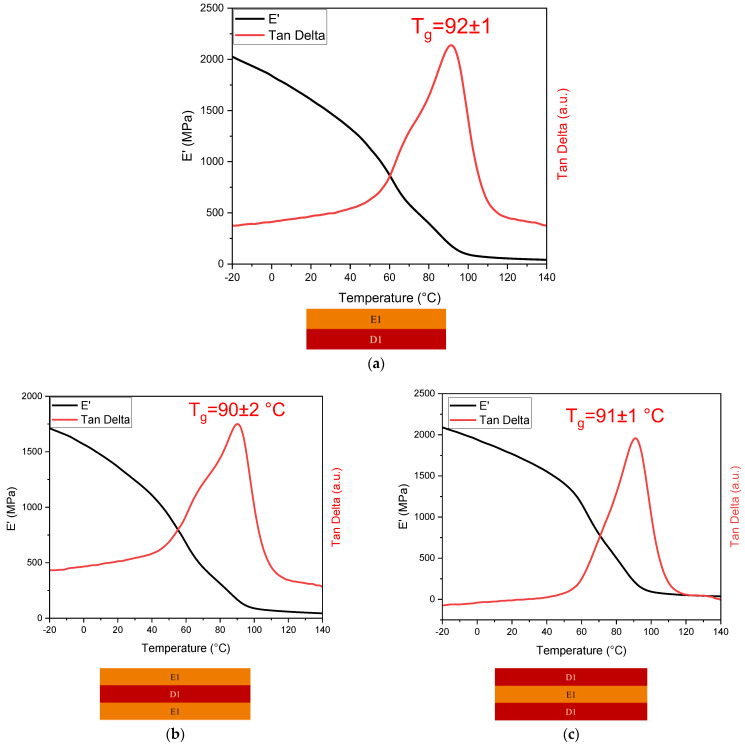
DMTA analyses of multi-material 3D printed samples: double-layer E1-D1 (**a**), triple-layer E1-D1-E1 (**b**), and triple-layer D1-E1-D1 (**c**).

**Figure 11 polymers-16-01510-f011:**
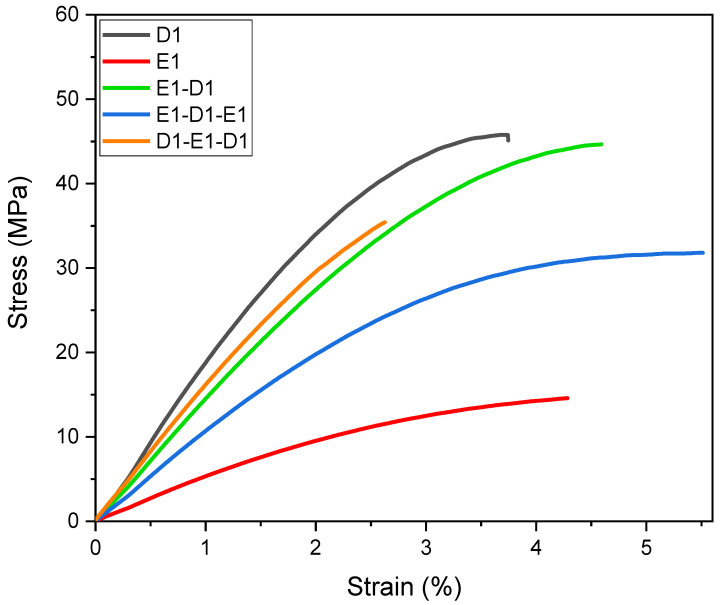
Stress–strain curves for different multi-material and mono material samples.

**Figure 12 polymers-16-01510-f012:**
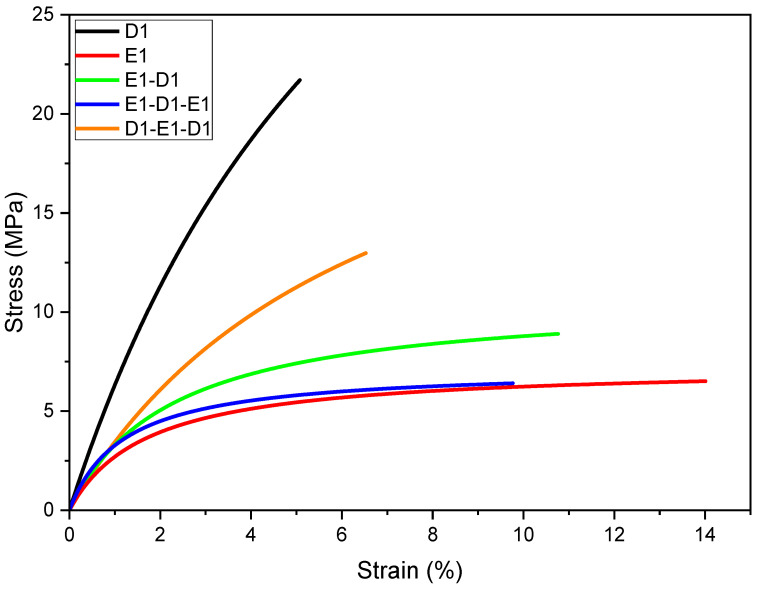
Stress–strain curves for different multi-material and single-material samples.

**Figure 13 polymers-16-01510-f013:**
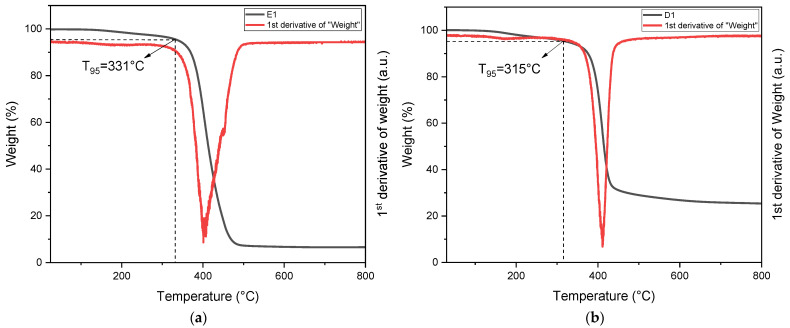
Thermal degradation of cured E1 resin (**a**) and D1 resin (**b**) obtained by TGA.

**Figure 14 polymers-16-01510-f014:**
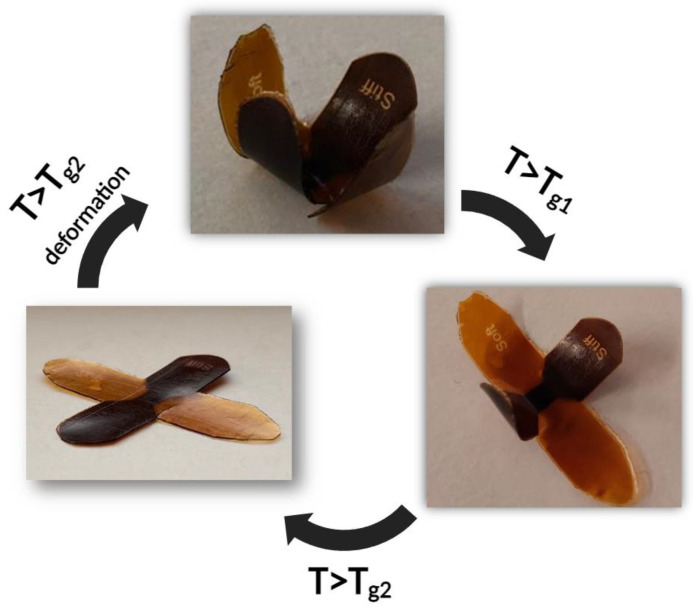
Different shape of the printed flower at different temperatures.

**Table 1 polymers-16-01510-t001:** Resin mixture composition.

Formulation ID	Biobased Monomer	Photoinitiator (phr)	ITX (phr)	Sudan II (phr)
E1	ELO	2	From 0 to 2	0.02
D1	DGEVA	2	From 0 to 2	0.02

**Table 2 polymers-16-01510-t002:** Photo-DSC data of E1 and D1 formulation with different ITX amounts obtained by irradiation at 405 nm and 50 °C.

E1	Peak (s)	Heat Released (mW/mg)	Area (J/g)
Pristine	87 ± 2	0.2 ± 0.0	54 ± 5
1ITX	16 ± 1	6.7 ± 0.3	224 ± 2
2ITX	15 ± 2	6.8 ± 0.14	231 ± 12
D1			
Pristine	84 ± 8	2.0 ± 0.0	487 ± 2
1ITX	18 ± 0	7.0 ± 0.2	457 ± 31
2ITX	12 ± 0	5.0 ± 0.8	415 ± 10

**Table 3 polymers-16-01510-t003:** Glass transition temperature of E1, D1, and multilayer materials.

Sample	Glass Transition Temperature T_g_ (°C)
E1	52 ± 2
D1	96 ± 0
E1-D1 double-layer	92 ± 1
E1-D1-E1 triple-layer	90 ± 0
D1-E1-D1 triple-layer	91 ± 1

**Table 4 polymers-16-01510-t004:** Mechanical properties obtained by tensile tests.

	Young’s Modulus (MPa)	Tensile Strength (MPa)	Elongation at Break (%)
E1	505 ± 46	15.4 ± 1.2	4.7 ± 0.7
D1	1615 ± 132	33.3 ± 7.0	2.7 ± 0.7
E1-D1 double-layer	1312 ± 102	33.2 ± 10.9	3.0 ± 1.4
E1-D1-E1 triple-layer	1105 ± 96	32.3 ± 0.5	4.2 ± 1.2
D1-E1-D1 triple-layer	1589 ± 67	28.4 ± 5.4	1.9 ± 0.5

**Table 5 polymers-16-01510-t005:** Values obtained from the flexural tests.

	Flexural Modulus (MPa)	Flexural Strength (MPa)	Elongation at Break (%)
E1	523 ± 25	7.1 ± 0.5	15.1 ± 01.0
D1	1422 ± 331	18.4 ± 1.8	6.4 ± 1.4
E1-D1	490 ± 140	8.9 ± 0.1	12.2 ± 0.5
E1-D1-E1	523 ± 5	7.4 ± 1.0	11.6 ± 1.2
D1-E1-D1	1217 ± 250	13.6 ± 0.5	7.2 ± 1.1

## Data Availability

Data are contained within the article and Appendix A.

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
