# Peer review of "Multi-Material 3D Printing of Biobased Epoxy Resins"

_polymers, 2024, doi:10.3390/polym16111510_

Round 1

Reviewer 1 Report

Comments and Suggestions for Authors

Explain the reason behind using epoxy resins in 3D printing.

What is the role of AI in Additive manufacturing (AM)?

More explanation is required for Fig.3.

Figs.9&10 is missing. 

Explain the thermal degradation process.

Conclusions must be precise and attractive.

Comments on the Quality of English Language

minor

Author Response

The answer are reported in the attached file

Reviewer 2 Report

Comments and Suggestions for Authors

The issue of the Multi-material 3d printing of biobased epoxy resins is important in the domain. In this context, the presented work can be evaluated as modern and important to the field of additive manufacturing. However, the following concerns should be addressed to improve the quality and technical interpretation of the results.

1.      How and where this work can be implemented in Industries and at what level. Also mention this in the manuscript.

2.      Are presented values the result of single measurement? Have the measurements been replicated? What is the scatter of the results?

3.      Comment on the limitations of the present investigative study.

4.      Some background statement has to be added to the abstract.

5.      The introduction must be improved by adding details about the critical findings of other researchers. The discussion on vat polymerization, interpenetration networks, metamaterials, scaffolds, and epoxy is very important to be discussed for in-depth literature. All this information can be presented using the a) https://doi.org/10.1016/j.addma.2023.103607, b) 10.1021/acsami.3c06514  c) 10.1021/acsami.2c09602  d) 10.3390/polym14030566 e) https://doi.org/10.1016/j.cej.2022.141049 f) https://doi.org/10.1016/j.compscitech.2023.110357

6.      The conclusion must be restructured according to the objective of the current work.

7.      Photo curing process should be discussed thoroughly along with the behaviour of heat flow in Figure 3.

8.      Figure 9 and 10 is missing from the manuscript. Insert these images.

9.      Stress-strain curves for different multi-material and mono material samples should be discussed for all sample types.

10.   Avoid explanations in the conclusion section. Comment on the hypothesis of the study along with its validation. 

Comments on the Quality of English Language

Please correct minor typographical and grammatical errors by proofreading the manuscript. 

Author Response

The answer are rpeorted in the attached file

Reviewer 3 Report

Comments and Suggestions for Authors

Comments

In this paper, the authors applied the concept of dual-vat DLP 3D printing, which involves the automatic exchange of vats for the multi-material printing of bioderived epoxy resins. The theme is definite and the logic is clear. However, there are still some issues to be addressed. The specific comments can be found as following:

1.     For the writing of “3D” or “3d”, the title should be consistent with the full text.

2.     Attention should be paid to the format in which authors and their units are written, including superscript.

3.     Lines 65-67, for bioderived epoxy resins and their applications, you can refer to some of the latest literature, such as: Recent progress of biomass in conventional wood adhesives: a review.

4.     What are the bioderived epoxy resins other than ELO and DGEVA mentioned in the text? Do they have potential applications in multi-material 3D printing?

5.     Line 104, “photoinitiator p(octyloxyphenyl) phenyliodonium hexafluoroantimoniate” is inconsistent with the writing in Figure 1.

6.     Figures 9 and 10 are not shown in the manuscript.

7.     Table 3 and Table 5 have no serial number and name.

8.     There are still some typos and grammar issues in the manuscript. Authors should carefully recheck the whole manuscript.

Comments on the Quality of English Language

Need to be polished.

Author Response

The answer are reporete in the attached file

Round 2

Reviewer 2 Report

Comments and Suggestions for Authors

The authors have revised the manuscript as per the suggestions. Therefore, it can be accepted for publication. 

Author Response

We acknowledge the reviewer